# Pomological Descriptors, Phenolic Compounds, and Chemical Monitoring in Olive Fruits Irrigated with Dairy Treated Wastewater

**Wiem Sdiri [1], Samia Dabbou [2], Vincenzo Nava [3], Giuseppa Di Bella [3] and Hedi Ben Mansour [1,***

[1] Research Unit of Analysis and Process Applied on the Environment—APAE UR17ES32, Higher Institute of Applied Sciences and Technology, University of Monastir, 5000 Monastir, Tunisia; wiemsdiri@yahoo.fr

[2] Faculty of Dentistry of Monastir, University of Monastir, 5000 Monastir, Tunisia; samia.dabbou@fmdm.u-monastir.tn

[3] BioMorf Department, University of Messina, 98122 Messina, Italy; vnava@unime.it (V.N.); giuseppa.dibella@unime.it (G.D.B.)

* Correspondence: apae@issatmh.rnu.tn

**Abstract:** In this work, the pomological characteristics, phenolic composition, and chemical contents modification in response to treated wastewater (TWW) irrigation was studied on olive fruits. The experiment was carried out during two successive years (2016/2017) on olive trees (cv. Chemlali). Three irrigation treatments were adopted and two TWW irrigation levels were applied (T1: 20% $ET_c$; T2: 40% $ET_c$; CT: Control Treatment (rainfed condition)). Results show that TWW irrigation leads to increased fruit fresh weight and water content, whatever the level applied. In addition, fruit oil content remained unaffected by TWW irrigation. Moreover, this agronomic practice preserves some phenolic compound contents like verbascoside, therefore fruits nutritional value. A positive feature was then observed following TWW irrigation. In fact, oleuropein, tyrosol, luteolin-7-glucoside, and pinoresinol amounts were enhanced in treated olive fruits. On the other hand, TWW irrigated trees with a level of 40% $ET_c$ (T2) produced olive fruits richer in Mg and K than those cultivated in rainfed conditions (CT). Fruits Zn, Mn, and Pb contents decreased as a result of olive trees TWW irrigation.

**Keywords:** treated wastewater; irrigation; olive fruits; pomological characteristics; phenolic compounds; minerals

## 1. Introduction

In the Mediterranean Basin, water availability and low rainfall are a widespread problem. The scarcity of water resources is a limiting factor for economic development, particularly for the agriculture sector [1]. Therefore, the use of wastewater for irrigation purposes is one of the strategies adopted to alleviate the water shortage problem. The evaluation of wastewater quality is very important. In fact, this available resource can be classified as low-quality-water.

Food plants can be afflicted by inorganic contamination due to numerous factors including irrigation with wastewater as well as climate, atmospheric deposition, nature of the soil on which the plant is grown, and the application of fertilizers [2–4]. Inorganic elements may play an important role in the human body's metabolism. On the other hand, a poisonous effect can be observed when element intake exceeds the functional level. Iron, copper, nickel, and zinc in high concentrations are unbearable, while very low levels of mercury, lead, and cadmium have toxic effects [5].

The study of wastewater irrigation impact has been undertaken in many countries like Tunisia, where the main pillar of the economy is agriculture [6]. In Tunisia, olive growing is one of the main factors of social and economic stability in the country [7]. In olive growing areas, irrigation allows an increase in productivity and a reduction fruit drop. In fact, olive fruit's commercial value is strictly correlated with its satisfactory size, which

is obtained by a good water supply [8]. In contrary, irrigation with high salinity water decreases oil content, yield, and fruit weight [9]. Literature lacks information about the changes in pomological and biochemical characteristics of olive fruits following wastewater irrigation. Therefore, a significant development in technology, safety, and quality of table olives and olive oil require a better knowledge of this agronomic practice [7]. Wastewater has been used to irrigate many crops to help deal with water scarcity problems, especially in Mediterranean environments. In the case of olive orchard irrigation, varieties respond differently to this agronomic practice. In fact, more effort is needed in order to monitor some parameters such as toxic chemicals, which may pose a health risk for humans following olive fruit consumption. The effect of wastewater irrigation on fruit mineral elements has not been investigated. However, this aspect could be very important for olive orchard nutrient input and yield [10–12]. Thus, in this paper, we firstly studied the modification of fruit pomological traits (fruit fresh weight, pulp/stone ratio, fruit fresh stone weight, pulp percentage, fruit dry pulp weight, and oil content) in response to treated wastewater (TWW) irrigation. Then, we investigated the effect of TWW on fruit phenolic composition and mineral element content.

## 2. Material and Methods

### 2.1. Field Site and Plant Material

The experiment was conducted in 2016 and 2017 in an olive orchard of 160-year-old olive trees (*Olea europaea* L. cv. Chemlali) located in Dkhila, province of Mahdia, Tunisia (35°31′ N, 10°58′ E). This region is characterized by a Mediterranean climate. The mean annual temperature is 19.80 °C, and the mean annual rainfall is 348 mm. Climatic conditions have been described by Sdiri et al. [13], and the soil is sandy clay. The olive trees were spaced 14 m × 14 m apart (51 trees per ha). The trees were not treated with pesticides and had not been fertilized for years.

### 2.2. Experimental Design and Irrigation Schedule

As shown in Figure 1, the effect of wastewater irrigation on olive fruit quality was studied on a randomized block, which was split into three blocks on which three treatments were used. Each block contained five replicates. Complementary water irrigation was applied using treated wastewater (TWW). Trees grown under rainfed conditions correspond to the control treatment (CT), and trees receiving TWW with a seasonal water irrigation amount equivalent to 20% $ET_c$ and 40% $ET_c$ correspond to T1 and T2, respectively. These water irrigation adopted amounts were explained in our previous work [13]. Water was delivered monthly from March to May to avoid inflorescence fall, and every fifteen days from June to October for each year of study [13].

### 2.3. Water Irrigation Method

Treated wastewater was administered using a supply pipe, the method typically used by the farmers in the area of study. Each irrigation lasted 5 or 10 min corresponding to 20% $ET_c$ or 40% $ET_c$, respectively.

### 2.4. Industrial Treated Wastewater

The treated wastewater (TWW) used in this work was collected from a dairy industry wastewater treatment plant located in Mahdia City (central eastern Tunisia). The composition of this TWW was described in our previous study [14]. The TWW was devoid of pesticides, antibiotics, and heavy metals. TWW's physicochemical characteristics are within the limits established for TWW reuse in irrigation (NT 106.03).

### 2.5. Olive Fruit Sampling

At the end of the experiment (in December 2017), fruits from each tree were hand harvested, after which sub-samples of 2 kg of fruits were collected and transported immediately to the laboratory for analysis.

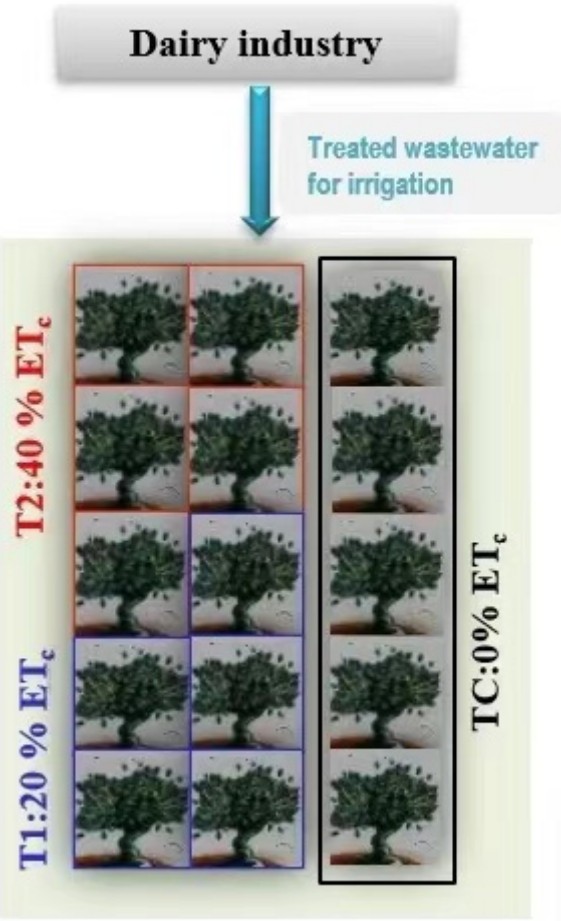

**Figure 1.** Experimental design diagram.

### 2.6. Fruits Pomological Analysis

Average fruit fresh weight (FFW) was determined for each treatment. Systematically, one hundred olive fruits were weighted in triplicate. Olives were de-stoned, the pulp and stone were weighed to determine the pulp to stone ratio (P/S), the pulp percentage (% pulp), and the average fruit fresh stone weight (FFSW). The water content was determined according to the procedure described by Chehab et al. [15]. According to this method, 100 olive fruits were dried in an oven at 70 °C until a constant weight was achieved. The oil content was then determined using Soxhlet apparatus (68 °C), with hexane as an extraction solvent.

### 2.7. HPLC Analysis of Phenolic Composition

Pulp phenols were separated by high performance liquid chromatography (HPLC). This process was carried out with a system containing an HP-1100 pump, a Rheodyne model 7725 injector (Cotati, CA, USA, loop volume 20 μL), a UV detector (280 nm), and a C18 Hypersil ODS (250 mm × 4 mm). The analytical conditions of fruit methanolic extracts were described by Tekaya et al. [16]. Phenolic compound identification was carried out by calculating and comparing retention times for each sample group with the corresponding standards and quantification by using cinnamic acid as an internal standard.

### 2.8. Element Determination by ICP-MS

Triplicate samples of each fruit (0.5 g) were digested as described by Sdiri et al. [13]. Fruit mineral composition was carried out using an Agilent 7500cx (Agilent Technologies, Santa Clara, CA, USA) inductively coupled plasma mass spectrometer (ICP-MS). The instrument was also equipped with an autosampler, ASX520 (Cetac Technologies Inc.,

Omaha, NE, USA), and an integrated sample introduction system. The ICP-MS operating conditions were those described by Di Bella et al. [17] and Potortì et al. [4].

### 2.9. Statistical Analysis of Data

The Duncan test was used to evaluate the effects of wastewater irrigation on olive fruits pomological descriptors, phenolic compounds, and element contents following the application of different treatments (T1, T2, and CT), using the statistical program SPSS 23.0 for Windows (SPSS, Chicago, IL, USA). All analyzed parameters were carried out in triplicate and the statistical significance level was fixed at $p < 0.05$.

## 3. Results

### 3.1. Changes in the Pomological Characteristics and in the Phenolic Composition of 'Chemlali' Olive Fruits Following TWW Irrigation

It can be seen in Table 1 that irrigated trees registered the highest mean value of fruit fresh weight and water content compared with the non-irrigated control. Moreover, no significant differences were observed between the two TWW levels. Table 1 also shows that TWW irrigation of both levels maintained very similar average fruit fresh stone weights, pulp percentages, and fruit dry pulp weights. Our results revealed that total phenol content in fruit coming from trees receiving 40% $ET_c$ (16,761.90 mg kg$^{-1}$) did not vary significantly compared to controls. On the other hand, changes to the amount of phenolic compounds gave different profiles according to the applied treatment (Table 2). The verbascoside content (the most abundant phenol) was not affected by TWW irrigation. In contrast, oleuropein and tyrosol amounts increased with TWW irrigation but behaved differently depending on the level. In fact, tyrosol concentration increased when the TWW level increased to T2, reaching a value of 539.90 mg kg$^{-1}$, whereas oleuropein concentration increased similarly following T1 and T2 application. With respect to CT, TWW application increases pinoresinol content by 428.40% and 590.36% following T1 and T2 exposure, respectively. Meanwhile, catechin-hydrate, 4HO-Benz, and luteolin-7-rutinoside amounts remained unaffected by TWW irrigation.

**Table 1.** Changes in fruit fresh weight, pulp/stone ratio, fruit fresh stone weight, pulp percentage, fruit dry pulp weight, oil content, water content, and phenol content after treatment with wastewater irrigation (T1 and T2).

| Parameters | CT | T1 | T2 |
|---|---|---|---|
| Fruit fresh weight (g) | 0.44 ± 0.02 [b] | 0.70 ± 0.00$_2$ [a] | 0.60 ± 0.09 [a] |
| Pulp/stone ratio | 1.14 ± 0.20 [b] | 1.92 ± 0.31 [a] | 2.00 ± 0.50 [a] |
| Fruit fresh stone weight (g) | 0.20$_8$ ± 0.00$_7$ [a] | 0.23 ± 0.02 [a] | 0.20 ± 0.01 [a] |
| Pulp percentage (%) | 53.22 ± 4.15 [a] | 65.50 ± 4.00 [a] | 65.60 ± 5.18 [a] |
| Fruit dry pulp weight (g) | 0.13 ± 0.02 [a] | 0.15 ± 0.03 [a] | 0.15 ± 0.02 [a] |
| Oil content (%) | 20.40 ± 5.01 [a] | 28.00 ± 2.30 [a] | 24.94 ± 0.90 [a] |
| Water content (%) | 32.41 ± 4.53 [b] | 42.53 ± 2.40 [a] | 43.40 ± 3.14 [a] |
| Phenol content (mg kg$^{-1}$) | 19,809.52 ± 400.54 [a] | 10,746.70 ± 158.30 [b] | 16,761.90 ± 337.60 [ab] |

CT = control treatment (trees grown under rainfed conditions); $ET_c$ = crop evapotranspiration; T1 = irrigation treatment 20% $ET_c$; T2 = irrigation treatment 40% $ET_c$. Data represents mean values ± standard deviation. Horizontally, values with the same letter are not significantly different at 5% probability level according to Duncan test.

### 3.2. Effect of TWW Irrigation on Olive Fruits Chemicals

Fruit element concentrations, reported in Table 3, demonstrate that P content is lower in fruit from TWW irrigated trees, especially following T1 application—after which the recorded level was 1240.58 mg kg$^{-1}$. The opposite trend was observed for K content. In fact, fruit K concentration increased following T2 application, reaching a value of 13,026.17 mg kg$^{-1}$. In addition, Table 3 reveals a significant enhancement in Na content in response to TWW use for irrigation, especially as a result of 20% $ET_c$ (T1) application. In contrast, T1 treatment led to a decrease in Ca concentration, which was 33.89% lower than the value observed for the control treatment (CT). Our results indicate (Table 3) a significant



decrease in terms of Zn and Mn accumulation in fruit coming from TWW irrigated trees, especially following T2 application, compared to amounts characterizing control fruit. These values decreased by 7.27% and 19.56% for T1 and T2, respectively. After 2 years of TWW irrigation, a significant decrease of Pb content was observed in fruit coming from trees receiving 20% $ET_c$ (T1).

**Table 2.** Contents of phenolic compounds (mg kg$^{-1}$) in olive fruit samples after treatment with wastewater irrigation.

| Compound | CT | T1 | T2 |
|---|---|---|---|
| Catechin-hydrate | 202.80 ± 30.70 [a] | 234.80 ± 20.50 [a] | 213.40 ± 12.15 [a] |
| Tyrosol | 293.10 ± 12.90 [b] | 255.05 ± 14.40 [b] | 539.90 ± 5.81 [a] |
| 4-Hydroxy-Benzoic acid | 126.80 ± 3.60 [a] | 157.90 ± 2.15 [a] | 140 ± 9.80 [a] |
| Luteolin-7-rutinoside | 204.70 ± 9.00 [a] | 134.70 ± 13.30 [a] | 145.70 ± 0.80 [a] |
| Verbascoside | 341.90 ± 24.40 [a] | 398.80 ± 18.08 [a] | 466.10 ± 14.50 [a] |
| Luteolin-7-glucoside | 145.80 ± 2.10 [b] | 203.30 ± 8.08 [b] | 391.30 ± 0.10 [a] |
| Apigenin-7-glucoside | 159.50 ± 9.30 [c] | 287.40 ± 1 [a] | 233.90 ± 3.05 [b] |
| Oleuropein | 306.80 ± 12.70 [b] | 797.20 ± 8.30 [a] | 741.80 ± 14.90 [a] |
| Pinoresinol | 30.10 ± 1.60 [b] | 159.05 ± 6.40 [a] | 207.80 ± 9.70 [a] |

CT = control treatment (trees grown under rainfed conditions); $ET_c$ = crop evapotranspiration; T1 = irrigation treatment 20% $ET_c$; T2 = irrigation treatment 40% $ET_c$. Data represents mean values ± standard deviation. Horizontally, values with the same letter are not significantly different at 5% probability level according to Duncan test.

**Table 3.** Mineral element content (mg kg$^{-1}$) of olive fruits after treatment with wastewater irrigation.

| Element | CT | T1 | T2 |
|---|---|---|---|
| Na | 1164.74 ± 0.00$_1$ [c] | 1732.19 ± 0.00$_3$ [a] | 1682.04 ± 0.00$_2$ [b] |
| Mg | 2095.91 ± 0.00$_1$ [b] | 1738.12 ± 0.00$_2$ [c] | 2102.94 ± 0.00$_1$ [a] |
| P | 1820.62 ± 0.00$_2$ [a] | 1240.60 ± 0.00$_1$ [c] | 1424.15 ± 0.00$_2$ [b] |
| K | 12,827.08 ± 0.01 [b] | 12,309.80 ± 0.00$_2$ [c] | 13,026.17 ± 0.00$_2$ [a] |
| Ca | 1551.51 ± 0.00$_3$ [a] | 1025.34 ± 0.00$_2$ [c] | 1354.40 ± 0.00$_1$ [b] |
| Mn | 18.40$_7$ ± 0.00$_1$ [a] | 12.82 ± 0.00$_3$ [c] | 14.80 ± 0.00$_7$ [b] |
| Fe | 199.70 ± 0.00$_2$ [a] | 51.92 ± 0.00$_3$ [c] | 74.70 ± 0.60 [b] |
| Cu | 43.80 ± 0.00$_3$ [b] | 52.30 ± 0.60 [a] | 32.53 ± 1.09 [c] |
| Zn | 50.80 ± 0.10 [a] | 42.83 ± 0.15 [c] | 47.01 ± 0.81 [b] |
| V | 0.70$_4$ ± 0.00$_5$ [a] | 0.51 ± 0.00$_2$ [c] | 0.60 ± 0.00$_5$ [b] |
| Cr | 0.70 ± 0.00$_5$ [a] | 0.08 ± 0.00$_5$ [c] | 0.20 ± 0.00$_2$ [b] |
| Co | 0.05 ± 0.00$_5$ [a] | 0.03 ± 0.01 [b] | 0.03 ± 0.00$_5$ [b] |
| Ni | 0.30 ± 0.00$_1$ [c] | 0.80$_6$ ± 0.00$_6$ [a] | 0.40 ± 0.00$_1$ [b] |
| As | 0.05 ± 0.00$_2$ [a] | 0.04 ± 0.00$_3$ [b] | 0.05 ± 0.00$_1$ [a] |
| Se | 0.05 ± 0.00$_2$ [b] | 0.03 ± 0.00$_1$ [c] | 0.08 ± 0.00$_4$ [a] |
| Cd | tr [a] | tr [a] | tr [a] |
| Hg | 0.07 ± 0.00$_1$ [a] | 0.05 ± 0.00$_1$ [b] | 0.05 ± 0.00$_1$ [b] |
| Pb | 0.44 ± 0.00$_2$ [a] | 0.30 ± 0.00$_1$ [c] | 0.40$_2$ ± 0.00$_1$ [b] |

CT = control treatment (trees grown under rainfed conditions); $ET_c$ = crop evapotranspiration; T1 = irrigation treatment 20% $ET_c$; T2 = irrigation treatment 40% $ET_c$. Data represents mean values ± standard deviation; tr = values < 0.01 (trace). Horizontally, values with the same letter are not significantly different at 5% probability level according to Duncan test.

## 4. Discussion

The results of the current study prove that TWW irrigation enhances fruit fresh weight, a finding that is in consonance with data published by Bourazanis et al. [18] regarding Koroneiki cultivar. The same result was observed in water content. These parameters are likely well correlated. This enhancement can be explained by water supplement by irrigation [19]. In addition, this result reflects good TWW quality. In fact, water stress can cause a drop in fruit water content and growth [8].

Irrigation with TWW did not significantly affect the oil content. In the work of Bourazanis et al. [18], however, oil content was influenced negatively following treated municipal wastewater irrigation. According to Seçmeler and Galanakis [20], the oil fraction was about 20% of the olive fruit composition. In the present data, this was true of control fruit, while wastewater irrigation induced an oil content slightly more than 20% of the olive fruit composition.

Our findings regarding total phenol content are in contrast with those obtained by Pedrero et al. [21], whose findings indicated that total phenol concentration was enhanced in nectarine fruits irrigated with tertiary-treated reclaimed wastewater. However, this water presents high salinity and likely causes saline stress, which activates the biosynthesis of phenols [21]. According to Patumi et al. [22] and Tovar et al. [23], phenol content decreased when the irrigation water applied increased. In our case, this parameter was maintained unchanged. After an increase in irrigation level to 40% $ET_c$, it can be seen that this TWW amount preserves good olive fruit quality.

Statistical analysis demonstrated that all phenolic compound content did not decrease following TWW irrigation. In a previous study by Tekaya et al. [7] on 'Chemlali' olive trees, wastewater caused a decline in the levels of most phenolic compounds. It can be concluded from our results, however, that TWW maintains olive fruit phenolic compound content, and consequently its nutritional value and beneficial effects on human health [7,24].

Oleuropein was accumulated mainly in high concentrations in fruit corresponding to blocks treated with T1 and T2. This accumulation suggests an increase in olive fruit bitter taste [7]. Moreover, this enhancement can be considered a positive feature because of oleuropein's several benefits for human health [25].

Fruits produced by olive trees irrigated with 40% $ET_c$ (T2) recorded an enhancement in terms of tyrosol and luteolin-7-glucoside content compared to those of control treatment (CT). Our results agree with the finding of Tekaya et al. [7], which reported a fruit tyrosol content increase following olive tree wastewater irrigation. In contrast, Patumi et al. [26] reported that irrigation induces a decrease in this phenolic compound content. In fact, it may be asserted that irrigation water type influences olive tyrosol concentration. The work of Tekaya et al. [7] demonstrated that olive tree wastewater irrigation decreases the luteolin-7-glucoside content of olive fruit.

The mean value found for verbascoside remained unchanged between rainfed conditions and TWW irrigation. This unchanged value may be related to the unaffected pulp percentage. In fact, phenolic compounds, such as verbascoside, are mainly concentrated in the pulp [27]. This type of water likely preserves fruit nutritional value, maintaining the clinical potential and biological properties granted by verbascoside [28].

The literature lacks information about the contribution of wastewater irrigation to olive fruit phenolic profile changes, but the work of Sdiri et al. [13] did not show any negative effect of this practice on oil pinoresinol content. This finding is in line with our data, which indicate an olive pinoresinol amount increase following treated wastewater irrigation.

In contrast with Ca content, fruit fresh weight enhanced as a result of T1 application compared to that of fruit produced under control conditions (CT). Our results are in accordance with those recorded by Kartas et al. [29], indicating a negative correlation between fruit weight and Ca content.

The significant P content decrease reported in fruits from TWW irrigated trees, especially following T1 application, may be the consequence of increased fruit fresh weight [12]. Fruit K content enhancement following T2 application can be explained by K accumulation for amino acid and protein synthesis [12].

Olive tree TWW irrigation with a level of 20% $ET_c$ (T1) leads to an increase in Na content in olive fruit. White and Broadley [30] found that this element was closely linked to water movement and uptake fluxes inside the tree. In the present work, this data may be confirmed by the fruit fresh weight and water content enhancement in fruit coming

from trees receiving 20% ET$_c$ (T1) as compared to values registered in fruits obtained from control treatment (CT).

Our results indicated a significant decrease in Zn, Mn, and Pb in fruit coming from TWW irrigated trees. As reported by Zaanouni et al. [6], this treatment likely promotes the mobilization of minerals that are potentially toxic to the leaves. It is well known that high concentrations of Zn, Mn, and Pb in olives can harm human health [6].

## 5. Conclusions

In arid and semi-arid regions, fundamental knowledge about the effects of treated wastewater use in irrigation should be improved to deal with the freshwater scarcity problem. This agronomic practice proved its efficiency in the present work by preserving olive fruit commercial value and improving its nutritional value. In fact, oil content was not negatively affected by TWW irrigation, and total phenol content was not increased. These data reflect good wastewater quality that did not present high salinity. Therefore, phenolic compound content (catechin-hydrate, verbascoside, etc.) was maintained to preserve the fruits nutritional value. The increase of some phenolic compound content like oleuropein and pinoresinol following treatment with wastewater irrigation can be considered as a positive feature because of their several benefits for human health. In our work, TWW irrigation contributed positively to an increase in fruit fresh weight and olive nutrient content like Mg and K levels.

Based on the results of this study, TWW can safely be recommended for olive tree irrigation, since this agronomic practice did not cause fruit contamination by potentially toxic elements like Zn, Mn, or Pb. Rather, TWW use for olive orchard irrigation seems to be advantageous by decreasing Zn, Cr, and Pb contents in olive fruit compared to results obtained in rainfed conditions.

**Author Contributions:** Methodology, writing—original draft preparation, W.S.; review and editing, S.D.; visualization, V.N.; formal analysis, G.D.B.; supervision, H.B.M. All authors have read and agreed to the published version of the manuscript.

**Funding:** This research received no external funding.

**Institutional Review Board Statement:** Not applicable.

**Informed Consent Statement:** Not applicable.

**Conflicts of Interest:** The authors declare no conflict of interest.

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
