# Peer review of "Pomological Descriptors, Phenolic Compounds, and Chemical Monitoring in Olive Fruits Irrigated with Dairy Treated Wastewater"

_chemosensors, doi:10.3390/chemosensors9060130_

Round 1

Reviewer 1 Report

  1. In Introduction section, the authors are suggested to provide more conclusive statements with related citations about the wastewater irrigation and its possible effects to olive trees or others. Present information is not well supported to the scope of the work.
  2. Water irrigation methods need to included in the material and methods section. 
  3. Adding relevant pictures/figures will add significance to the study.
  4. Why the phenols content reduced in the treatment compared to the control ?? Please support your findings and explanations.
  5. Provide some comparative discussion about the changes in the contents of phenolic compounds that were observed in your study. 
  6. Conclusions section can be improved by adding important findings of the results by mentioning  the significant increase of fruit weight, changes in phenol content and other parameters influencing the fruit characteristics. 

Reviewer 2 Report

The paper presents results from studies on very important issue - use of wastewater for irrigation purposes in arid and semi-arid regions - with the aim to alleviate the water shortage problem. Really positive result from the work is the recommendation to use the irrigation with studied treated wastewater as sure means to cope with water problem in the region without compromising the quality of olive fruits. Interesting results with respect to different parameters describing the fruits quality are presented and more important - an attempt is made these results to be explained.

The materials and methods used are accurately and clearly described. Perhaps the availability of a little more information about the composition of the treated wastewater used for irrigation (and especially the content of nitrogen and phosphorus) would be useful to better understand the results.

The abstract and conclusion correctly present and summarize the work done and results obtained.

Some corrections in the English are needed, some expressions have to be clarified. For example, from the statement (page 3)

''It was found in Table 2 that, irrigated trees registered the highest mean values of fruit fresh weight, P/S ratio, and water content compared to controls non-irrigated. Moreover, we didn’t mark significant differences between TWW levels. The same Table showed that  TWW irrigation whatever the level maintained unchanged average fruit fresh stone weight, pulp percentage and fruit dry pulp weight''

the reader understands that the pulp percentage is unchanged for TWW irrigated fruits.

From the statement (page 5)

''The mean value found for verbascoside remained unaffected passing from rainfed conditions to TWW irrigation. This unchanged value may be referred to the unaffected  pulp percentage.''

the reader understands that the pulp percentage is unchanged for the three types of  irrigated fruits.

Somewhere coordination of verb tenses is needed, fore example:

''Results showed that TWW irrigation  leads to increase fruit fresh weight and water content whatever the level applied.''

Results have shown that TWW irrigation lead to increase fruit fresh weight and water content whatever the level applied.

or

Results show that TWW irrigation leads to increase fruit fresh weight and water content whatever the level applied.

As an overall recommendation the paper can be accepted after minor revisions related to the comments above and in the paper file that I am sending to make the authors work easier.
